# Peripheral mitochondrial DNA, telomere length and DNA methylation as predictors of live birth in *in vitro* fertilization cycles

Letizia Li Piani[1,2]*, Marco Reschini[2], Edgardo Somigliana[1,2], Stefania Ferrari[2], Andrea Busnelli[3,4], Paola Viganò[2], Chiara Favero[1], Benedetta Albetti[1], Mirjam Hoxha[1], Valentina Bollati[1]

**1** Dept of Clinical Sciences and Community Health, Università degli Studi di Milano, Milan, Italy, **2** Infertility Unit, Fondazione IRCCS Ca' Granda Ospedale Maggiore Policlinico, Milan, Italy, **3** Department of Biomedical Sciences, Humanitas University, Pieve Emanuele, Milan, Italy, **4** Division of Gynecology and Reproductive Medicine, Department of Gynecology, IRCCS Humanitas Research Hospital, Fertility Center, Rozzano, Milan, Italy

* letizia.lipiani@unimi.it

**Data Availability Statement:** All relevant data are within the paper and its Supporting Information files.

## Abstract

### Objective

To evaluate whether telomere length (TL), mitochondrial-DNA (mt-DNA) or epigenetic age estimators based on DNA methylation (DNAm) pattern could be considered reliable predictors of in-vitro-fertilization (IVF) success in terms of live birth rate.

### Design

Prospective cohort study

### Setting

Infertility Unit of the Fondazione IRCCS Ca' Granda Ospedale Maggiore Policlinico

### Patients

181 women aged 37–39 years who underwent IVF at a single-centre between January 2017 and December 2018.

### Interventions

On the day of recruitment, blood samples were collected, and genomic DNA was isolated from white blood cells. TL, mt-DNA and DNAm assessment was performed using quantitative real-time polymerase chain reaction (qPCR). Biological age (DNAm age) was computed as the algorithm based on methylation pattern of five genes. Epigenetic age acceleration was estimated from the residuals of the linear model of epigenetic age regressed on chronological age. Long Interspersed Nuclear Elements (LINE)-1 methylation pattern was used as a surrogate for global DNA methylation.

**Funding:** The author(s) received no specific funding for this work.

**Competing interests:** I have read the journal's policy and the authors of this manuscript have the following competing interests: Dr. Somigliana reports grants from Ferring, grants and personal fees from Merck-Serono, grants and personal fees from Theramex, personal fees from Gedeon-Richter, outside the submitted work. This does not alter our adherence to PLOS ONE policies on sharing data and materials, as detailed online in your guide for authors. All the other authors have no competing interests in relation to this study.

## Main outcome measures

This study investigated whether peripheral TL, mt-DNA and DNAm could predict live birth in IVF cycles.

## Results

TL, mt-DNA and LINE-1 methylation were not associated with IVF success. Conversely, DNAm age resulted significantly lower in women who had a live birth compared to women who did not (36.1 ± 4.2 and 37.3 ± 3.3 years, respectively, p = 0.04). For DNAm age, odds ratio (OR) for live birth per year of age was 0.90 (95%CI: 0.82–0.99, p = 0.036) after adjusting for FSH and antral follicle count (AFC) and 0.90 (95%CI: 0.82–0.99, p = 0.028) after adjusting also for number of oocytes retrieved. A significant association also emerged for epigenetic age acceleration after adjustments (OR = 0.91, 95%CI: 0.83–1.00, p = 0.048).

## Conclusion

DNAm age is associated with IVF success but the magnitude of this association is insufficient to claim a clinical use. However, our findings are promising and warrant further investigation. Assessment of biological age using different epigenetic clocks or focusing on different tissues may reveal new predictors of IVF success.

## Introduction

Assisted reproductive technology (ART) has spread worldwide in recent years. Nevertheless, the chances of pregnancy per cycle have only marginally improved. According to the latest report of the European registers, the clinical pregnancy rate per cycle is only 28% [1]. Age and ovarian reserve are crucial determinants of success but, even in the best prognosis group, the chances of failure prevail. There is a general consensus that the high rate of aneuploidy observed in embryos derived from in-vitro fertilization (IVF) procedures plays a central role and could explain the sharp decrease of success with age [2,3]. Aging, however, is not a uniform process and may affect each one differently. In other words, chronological and biological age may not completely overlap. Investigating whether peripheral biomarkers of aging are associated with the IVF success could provide an appealing new tool for counselling and screening. In this scenario, we hypothesised mitochondrial DNA, telomere length and DNA-methylation pattern as possible useful predictors.

Mitochondrial DNA (mt-DNA) represents a surrogate measurement of the concentration of mitochondria per cell. It reflects the bioenergetics potential of the cells and its concentration was associated with aging and infertility [4]. To note, a previous study investigating the relation between mt-DNA and ovarian reserve highlighted a positive correlation between the two [5]. In addition, the authors observed a positive correlation between mt-DNA in peripheral blood cells and granulosa cells, suggesting that peripheral assessments could indirectly reflect the local situation within the ovary. Of further interest here is that, in natural conceptions, time to pregnancy may be longer in women with a lower peripheral mt-DNA copy number in peripheral blood [6].

Telomeres (derived from the Greek nouns telos, "end" and meros, "part") consist of < 5–15 kb-long tandem repeat hexanucleotide sequence $(TTAGGG)_n$ at the end of chromosomes, that

prevent genome from degradation, unwanted recombination and altered gene expression [7]. Telomere length (TL) decreases with age and its attrition is associated with age-related disorders, such as cardiovascular disease, diabetes, cancer [8–10]. There is a growing body of literature looking into a possible role of telomere length in reproductive aging. According to the telomere-based theory of reproductive aging, TL would represent a biomarker for biological aging and ovarian reserve [11–14].

Epigenetic clocks based on DNA methylation biomarkers were developed to enable estimates of biological age (DNAm age) [15]. Up to now, research has registered multiple models to assess DNAm age, different in terms of number and nature of the involved genes [15–17]. Indeed, the DNAm age estimators include multiple age-related methylation sites to define epigenetic metrics stably applicable across populations and then contrasted with chronological age to obtain epigenetic age acceleration estimates [15]. Epigenetic age acceleration understates a condition of frailty that could be used for the risk assessment of several diseases, from cancer to frailty [18,19]. It seems to be heritable [20] and, remarkably, genome-wide association studies have recently associated epigenetic age acceleration to telomerase reverse transcriptase, suggesting a bond between telomere biology and epigenetic clock [21]. On the other hand, the assessment of the methylation status of Long Interspersed Nuclear Elements-1 (LINE) is commonly used as a surrogate to measure global DNA methylation due to their high occurrence throughout the genome [22–24].

In this study, we aimed at disentangling the possible association of peripheral determinants of aging and IVF success. To this aim, we set-up a prospective cohort study in women in the late thirties undergoing IVF cycles and compare mt-DNA, TL and DNAm between those who did and did not have a live birth.

## Materials and methods

Women referred to the Infertility Unit of the Fondazione IRCCS Ca' Granda Ospedale Maggiore Policlinico and scheduled for IVF or Intracytoplasmic Sperm Injection (ICSI) between January 2017 and December 2018 were prospectively evaluated for study entry. Inclusion criteria were as follows: 1) indication to IVF, 2) age 37–39 years, 3) body mass index (BMI) 17–35 Kg/m$^2$, 4) serum FSH < 15 IU/mL, 5) less than 3 previous IVF cycles. Criteria for exclusion were as follows: 1) uterine abnormalities such as fibroids, adenomyosis, endometrial polyps and uterine septum, 2) hydrosalpinx at basal ultrasound, 3) severe male factor infertility (azoospermia requiring surgical recover of spermatozoa, cryptozoospermia, necrospermia, globozoospermia), 4) systemic diseases that could affect pregnancy outcome and cause miscarriage (such as diabetes, uncompensated thyroid disease, anti-phospholipid antibody syndrome). In addition, we excluded women who did not initiate the IVF cycle for any reason. Eligible subjects were invited to participate and signed an informed written consent prior to be recruited. The study was approved by the local Ethical Committee (*Comitato Etico Milano area B*, 2016/2176).

Recruited women provided a blood sample prior to initiate the IVF cycle. They were collected in EDTA-containing tubes and immediately stored at -80˚ C until assayed. Baseline clinical characteristics and IVF outcome of the selected women were obtained from patients' charts. Missed information were obtained by direct contact. An active investigation of the IVF cycle outcome was performed by phone call or by consulting patients' charts of the obstetrical unit of our hospital.

Molecular analyses were performed simultaneously after completing patients' recruitment and follow-up. Blood samples were thawed and DNA extracted using commercial kits as reported in details elsewhere [5].

Telomere Length (TL) and mt-DNA copy number (mt-DNAcn) were measured by using the real-time quantitative PCR method as described by Cawthon [25] and Hou et al. [26]. These assays measure relative TL and relative mt-DNAcn in DNA by determining, respectively, the ratio of telomere repeat copy number (T) and mitochondrial (mt) copy number to a single nuclear copy gene (S), which was the human (beta) globin (*hbg*). The T/S ratio and mt/S ratio are calculated in a given sample relative to a reference pool DNA. The reference pool DNA was prepared from 10 participants randomly selected from this same study, using 6 *μg* for each sample. A fresh 7-points standard curve prepared from the pooled DNA, ranging from 40 ng/*μ*l to 0.62 ng/*μ*l (serial dilutions 1 : 2), was included in every "T," "mt," and "S" PCR runs. For each sample, 9 ng of DNA was used as a template, and the reaction was run in triplicate. A high-precision MICROLAB STARlet Robot (*Hamilton Life Science Robotics*, *Bonaduz AG*, *Switzerland*) was used for transferring a volume of 7 *μ*l reaction mix and 3 *μ*l DNA (3 ng/*μ*l) in a 384-well format plate. All PCRs were performed on a 7900HT Fast Real-Time PCR System (Applied Biosystems). Primers and termal cycles were previously reported [26]). At the end of each real-time PCR reaction, a melting curve was added to confirm the amplification specificity and the absence of primer dimers. The average of the three T and three mt measurements were divided by the average of the three S measurements to, respectively, calculate the T/S or the mt/S ratio for each sample. The coefficients of variation for mt-DNAcn and for TL were respectively 4.3% and 5.9%.

DNAm age was calculated considering the methylation pattern of 5 CpG sites at five genes (ELOVL2, C1orf132/MIR29B2C, FHL2, KLF14, TRIM59) as reported elsewhere [16]. The DNA samples (500 ng) were plated at a concentration of 25 ng/μL in plates of 96 wells each and were treated with sodium bisulfite using the EZ-96 DNA Methylation-Gold™ Kit (*Zymo Research; Irvine*, *CA*, *USA*) following the manufacturer's instructions and eluted in 200 μL. 10 μL of bisulfite-treated template DNA were added to 25 μL of GoTaq Hot Start Green Master mix (Promega), 1 μL of forward primer (10 μM), and 1 μL of 5′ end-biotinylated reverse primer (10 μM) to set up a 50 μL PCR reaction. PCR cycling conditions and primer sequences have been previously reported [16].

Biological age (Y) was calculated as follows:

$$Y = 8.052 + 55.673*ELOVL2 + 47.141*FHL2 + 62.870*KLF14 -$$
$$-29.075*MIR2B29B2C + 41.281*TRIM59$$

Epigenetic age acceleration was defined as the residuals of DNAm age regressed on chronological age.

To assess DNA methylation of LINE-1, bisulfite-PCR was performed with the following primers: forward 5'-`TTTTGAGTTAGGTGTGGGATATA`-3', reverse 5'-biotin–`AAATCAAAAA ATTCCCTTTC`-3' and sequencing primer 5'-`AGTTAGGTGTGGGATATAGT`-3' as previously reported [27].

The sample size was calculated for mt-DNA. Based on previous studies of our group on this variable [5,6], we estimated that at least 170 women (of whom one third achieving live birth) were necessary to demonstrate a mean difference of 15% with type I and II errors set at 0.05 and 0.20. Given the sub-optimal reproducibility of the assessment of mt-DNA in our experience, we were not stringent on the sample size and opted a priori for a slightly larger recruitment (up to 200). Baseline characteristics and ART outcomes of two groups were compared according to Student's *t* test, Mann-Whitney and Fisher's Exact test, as appropriate. P values below 0.05 were considered statistically significant. In case of significant differences in baseline characteristics, a multivariate logistic regression model was used to obtain adjusted measures of association between the chances of live birth and the studied peripheral biomarkers. More

specifically, we first performed a multivariate analysis for the whole cohort that took into account only baseline variables found to differ between the study groups (i.e., serum FSH and AFC). Then, we performed a second multivariate analysis focusing only on women who retrieved oocytes and adjusting for the baseline variables included in the first model as well as variables of the cycles up to the number of available oocytes (including therefore information on ovarian hyperstimulation and oocytes retrieved). Specifically, the model included serum FSH, AFC and number of oocytes retrieved. The aim of this second analysis was to better extract pure biological effects, thus excluding women who could not provide oocytes.

## Results

One-hundred ninety-one women were initially recruited. Ten of them did not initiate the IVF cycle for personal reasons, leaving 181 women for data analyses. Overall, 58 women (32% of our cohort) obtained a live birth. An overview of the baseline clinical characteristics of the whole cohort as well as of the comparisons between women who did and did not deliver are shown in Table 1.

Statistically significant differences emerged for AFC and serum FSH between the two groups. Table 2 illustrates the IVF cycle outcomes, again for the whole cohort and separately for women who did and did not achieve a live birth. Statistically significant differences emerged for number of oocytes retrieved, number of suitable oocytes, fertilization rate and number of available embryos at cleavage stage.

Mt-DNA, TL, LINE-1 methylation, biological age and epigenetic age acceleration in women who did and did not obtain a live birth are shown in Table 3. A statistically significant

**Table 1. Baseline clinical characteristics of the whole cohort and comparison between women who did and did not obtain a live birth.**

| Characteristics | All cohort n = 181 | Live birth n = 58 | NO live birth n = 123 | p |
|---|---|---|---|---|
| Age (years) | 37.9 ± 0.8 | 37.8 ± 0.8 | 38.0 ± 0.9 | 0.22 |
| BMI (Kg/m$^2$) | 22.4 ± 3.6 | 22.3 ± 3.4 | 22.4 ± 3.7 | 0.99 |
| Previous deliveries | 34 (19%) | 10 (17%) | 24 (20%) | 0.84 |
| AMH (ng/ml) | 2.8 ± 2.4 | 3.2 ± 2.7 | 2.6 ± 2.3 | 0.21 |
| AFC | 12 ± 7 | 15 ± 8 | 10 ± 6 | <0.001 |
| FSH (IU/mL) | 7.4 ± 2.2 | 6.9 ± 2.1 | 7.7 ± 2.3 | 0.02 |
| Duration of infertility (years) | 3.7 ± 2.6 | 4.1 ± 3.2 | 3.5 ± 2.3 | 0.21 |
| Smoke | 33 (18%) | 10 (17%) | 23 (19%) | 1.00 |
| Previous IVF cycles | | | | 0.33 |
| None | 136 (75%) | 42 (73%) | 94 (76%) | |
| One | 31 (17%) | 13 (22%) | 18 (15%) | |
| Two | 14 (8%) | 3 (5%) | 11 (9%) | |
| Indication to IVF | | | | 0.90 |
| Unexplained | 55 (30%) | 15 (26%) | 40 (33%) | |
| Endometriosis | 33 (18%) | 10 (17%) | 23 (19%) | |
| Anovulatory | 10 (6%) | 4 (7%) | 6 (5%) | |
| Tubal factor | 26 (14%) | 8 (14%) | 18 (15%) | |
| Male factor | 48 (27%) | 18 (31%) | 30 (24%) | |
| Mixed | 9 (5%) | 3 (5%) | 6 (5%) | |

AFC: Antral Follicle Count. AMH: Anti-mullerian hormone.

Data are reported as mean ± SD or number (percentage).

**Table 2. IVF cycle outcome in the whole cohort and comparison between women who did and did not obtain a live birth.**

| Characteristics | All cohort n = 181 | Live birth | NO live birth | p |
|---|---|---|---|---|
| | | n = 58 | n = 123 | |
| Protocol of hyperstimulation | | | | 0.11 |
| GnRh Antagonist | 148 (82%) | 45 (78) | 103 (84%) | |
| Long Protocol | 22 (12%) | 11 (19%) | 11 (9%) | |
| Flare up | 11 (6%) | 2 (3%) | 9 (7%) | |
| Cancelled cycles | 7 (4%) | 0 (0%) | 7 (6%) | 0.10 |
| Total dose of gonadotropins (IU) [a] | 2,001 ± 671 | 1,864 ± 626 | 2,070 ± 684 | 0.06 |
| Duration of stimulation (days) [a] | 8.8 ± 1.8 | 9.0 ± 1.9 | 8.7 ± 1.7 | 0.34 |
| N. of oocytes retrieved [a] | 7.6 ± 5.5 | 9.5 ± 6.0 | 6.7 ± 5.1 | 0.001 |
| No oocytes retrieved [b] | 2 (1%) | 0 (0%) | 2 (2%) | 0.55 |
| N. suitable oocytes [a] | 6.3 ± 4.4 | 7.9 ± 4.6 | 5.5 ± 4.1 | 0.001 |
| N. of women without suitable oocytes [a,c] | 2 (1%) | 0 (0%) | 2 (2%) | 0.55 |
| Technique [d] | | | | 0.08 |
| Conventional IVF | 88 (52%) | 36 (62%) | 52 (46%) | |
| ICSI | 82 (48%) | 22 (38%) | 60 (54%) | |
| Fertilization rate (%) [d] | 71 [50–93] | 75 [63–100] | 67 [50–85] | 0.009 |
| No embryos available for transfer [d] | 9 (5%) | 0 (0%) | 9 (8%) | 0.03 |
| N. cleavage stage embryos (72 h) [e] | 3.6 ± 2.4 | 4.8 ± 2.4 | 2.9 ± 2.1 | < 0.001 |

Data are reported as mean ± SD or median [interquartile range] or number (percentage).

LB: Live birth. IU: International Units, IVF: In vitro fertilization. ICSI: Intracytoplasmatic Sperm Injection; 2PN fertilized oocyte.

[a] Refers to women who were not cancelled (n = 174).

[b] Data refer to number of women without oocytes at retrieval.

[c] Suitable oocytes refer to metaphase II oocytes and type 1 cumulus-oocyte complex according to the European Society for Human Reproduction and Embryology Istanbul Consensus Conference, 2011.

[d] Data refer to subjects retrieving at least one suitable oocyte (n = 170).

[e] The percentages refer to the number of women with available embryos (n = 161).

difference emerged only for DNAm age. Specifically, the mean ± SD biological age was 36.1 ± 4.2 and 37.3 ± 3.3 years respectively (p = 0.04) for women who did and did not have a live birth. The ROC curve aimed at assessing the accuracy of DNAm age in predicting live birth rate showed an AUC of 0.57 (95%CI: 0.48–0.66). Women with a biological age ≤ 10th centile (corresponding to 32 years) had an Odds Ratio (OR) of 2.4 (95%CI: 1.0–5.9, p = 0.05) for live birth.

Multivariate analyses adjusting for the variables found to differ between women who did and did not achieve a live birth are shown in Table 4. As aforementioned before, two different

**Table 3. Biomarkers of aging in women who did and did not achieve a live birth.**

| Variable | Live birth | NO live birth | p |
|---|---|---|---|
| | n = 58 | n = 123 | |
| Mitochondrial DNA (copy number) | 1.01 ± 0.27 | 1.04 ± 0.30 | 0.47 |
| Telomere length (TTAGGG repeats) | 1.04 ± 0.23 | 1.00 ± 0.27 | 0.32 |
| LINE-1 methylation (%5mC) | 76.4 ± 1.4 | 76.5 ± 1.5 | 0.71 |
| DNAm age (years) | 36.1 ± 4.2 | 37.3 ± 3.3 | 0.04 |
| Age acceleration (years) | 0.66 ± 4.0 | 0.34 ± 3.3 | 0.08 |

Data are reported as mean ± SD; p: p-value; DNAm age: Biological age.

**Table 4. Univariate and multivariate analyses on the relation between biomarkers of aging and live birth.**

| Variable | Univariate analysis | | | Model 1 * | | | Model 2 ** | | |
|---|---|---|---|---|---|---|---|---|---|
| | Crude OR | 95% CI | p | Adj. OR | 95% CI | p | Adj. OR | 95% CI | p |
| Mitochondrial DNA (copy number) | 0.67 | 0.22–2.00 | 0.47 | 0.94 | 0.29–3.03 | 0.91 | 0.95 | 0.29–3.07 | 0.93 |
| Telomere length (TTAGGG repeats) | 1.81 | 0.55–5.96 | 0.33 | 1.44 | 0.41–5.01 | 0.57 | 1.54 | 0.44–5.43 | 0.50 |
| LINE-1 methylation (%5mC) | 0.96 | 0.78–1.18 | 0.71 | 0.97 | 0.78–1.22 | 0.80 | 0.94 | 0.75–1.19 | 0.61 |
| DNAm age (years) | 0.91 | 0.84–1.00 | 0.048 | 0.90 | 0.82–0.99 | 0.036 | 0.90 | 0.82–0.99 | 0.028 |
| Age acceleration (years) | 0.92 | 0.84–1.01 | 0.08 | 0.91 | 0.83–1.00 | 0.06 | 0.91 | 0.83–1.00 | 0.048 |

* Data was adjusted for AFC and FSH using a multivariate logistic regression model.

** Data was adjusted for AFC, FSH and oocytes retrieved using a multivariate logistic regression model.

AFC: Antral Follicle Count. MtDNA: Mitochondrial DNA.

OR: Odds ratio; CI: Confidence interval; p: p-value.

ORs refer to the chance of live birth.

models were used for adjusting the data. Using this model, only DNAm age was found to differ between the study groups. The adjusted OR of live birth per year of age was 0.90 (95%CI: 0.82–0.99, p = 0.036). This finding suggested that successful pregnancy regarded more likely epigenetically younger women, according to their methylation pattern. In the second model, based on both baseline and IVF variables, a statistically significant difference persisted for DNAm age. The adjusted OR of live birth per year of age was 0.90 (95%CI: 0.82–0.99, p = 0.028). In this case, the difference in DNAm age between the two groups was not affected by the most common markers of ovarian reserve. In addition, a significant association emerged for epigenetic age acceleration. The adjusted OR of live birth per year was 0.91 (95%CI: 0.83–1.00, p = 0.048).

Finally, we performed a subgroup analysis based on the leading cause of infertility. Only two groups (endometriosis and mixed factor) were characterized by a statistically significant difference of biological age and age acceleration between women who delivered and who did not (p value: 0.03 and 0.04 respectively) (S1 Table). Besides, the difference in biological age persisted even comparing endometriosis to the unexplained cause of infertility, that could be considered our internal control group (p value: 0.04) (data not shown).

## Discussion

In this study, we failed to show a marked association between mt-DNA, TL, LINE-1 methylation, and the chances of live birth in IVF. In contrast, a statistically significant difference emerged for DNAm age. Given the modest difference, the clinical relevance is however doubtful. The area under the ROC curve indicates a low performance (0.57, 95%CI: 0.48–0.66). An exploratory analysis aimed at disentangling whether this biomarker could be more reliable to predict the success or the failure tended to favour the former (the OR for a biological $\leq 10^{th}$ centile being 2.4, 95%CI: 1.0–5.9) but additional evidence is needed.

The absence of correlation between mt-DNA and IVF success stands in apparent contrast to the current state of research. Bonomi et al. noted a biological grading of mt-DNA copy number in peripheral blood along with ovarian reserve, highlighting the poorest mt-DNA content in case of premature ovarian aging [5]. However, it must be underlined that oocytes quality and quantity are two sides of the same *coin*, that is female fertility [28,29]. They both decrease with age, but they have a different and independent trajectory. The contrast of our findings with those reported by Busnelli et al. is more difficult to reconcile [6]. Indeed, in that

study, mt-DNA correlated with time to natural pregnancy, indirectly suggesting a benefit on oocytes quality rather than on ovarian reserve. Comparison of our results with those of Busnelli et al. may be difficult as they are drawn from two different populations. While we focused on infertile women who referred to an infertility centre, Busnelli et al. studied pregnant women, that were defined subfertile for the length of pregnancy seek. Again, it would be simplistic to assimilate natural and IVF pregnancies. The occurrence of natural pregnancies implies a plethora of other physiological mechanisms that are not involved in IVF pregnancies (including ovulation, tubal function, uterine motility). Moreover, as Busnelli et al. underlined, pregnancy itself could impair circulating mtDNA, so that the comparison with our research may be not adequate.

Considering TL, the literature is more ample but also more conflicting. The implication of the telomere pathway in fertility is supported both by animal and human models [12,30]. Keefe et al. hypothesised that progressive shortening of telomeres, from foetal oogenesis to the adult ovary, would be the cause of ovarian impairment related to age [12]. Moreover, in case of ovarian insufficiency, shorter telomeres have been registered in leukocytes and granulosa cells [13]. On the other hand, it has recently been supposed that the maintenance system of TL may be atypical in the ovary. Both Morin et al. [31] and Lara-Molina et al. [32] found out that telomere length is higher in cumulus cells rather than in peripheral blood leukocytes, suggesting that the follicular environment could possess peculiar mechanisms to cope against telomere shortening. Similarly, Hanson et al. and Olsen et al have confirmed a difference in TL between cumulus or granulosa cells and white blood cells [14,33]. Based on their studies, Lara-Molina et al. stand against the use of TL of peripheral cells as a reliable indicator of follicular cell telomere length. These latest data fuel the idea that follicular telomere maintenance could not mirror other somatic tissue [34].

Despite the modest magnitude of the difference in DNAm age between women who did and did not deliver herein observed, evidence on this biomarker is the most promising and intriguing finding of our study. Based on a very small sample size, Monseur et al., 2020 observed a correlation between epigenetic clock and AMH and the number of oocytes yield, claiming the need to verify the parameter as an additional tool in ovarian reserve testing [35]. To note, in the present study, the association of DNAm age and the success of IVF was found to be independent from ovarian reserve biomarkers. It remained statistically significant even after adjusting for baseline characteristics (AFC and FSH) that sorted out to be statistically different among the two groups. Not only, but we could prove that the strength of the association between DNAm age and IVF success even increased adopting a more composite model for multivariate analysis. When adjusting for both baseline characteristics and the number of oocytes retrieved, we could appreciate that also epigenetic age acceleration turned out to be statistically significant. The possibility that epigenetic clock estimators could emerge as additional and independent predictors of ART success, assessing aspects different from ovarian reserve (oocyte retrieval, FSH and AFC), is particularly appealing. One could foresee a predictive algorithm taking into account ovarian reserve tests, chronological and DNAm age or epigenetic age acceleration. Besides, it could be hypothesised that specific diseases may sustain ovarian aging in infertile women. In our subgroup analysis, we noticed that women with endometriosis who could not have a live birth were biologically older than those who were successful.

There is a wide variety of models that can compute DNAm age and the most recent one relies on more than 30,000 genes [17]. We presently relied on a validated but simple method of DNAm age calculation based on the methylation of five genes. More comprehensive and potentially more precise approaches could be used. In addition, there is a recent trend to develop and validate surrogate clocks for clinical outcomes, including life span or age-related

diseases such as cancer and cardiovascular events rather than chronological age [17]. One may foresee the advance of clocks specifically dedicated to female fertility.

Although multi-tissue age estimators, such as the first one called Horvath's clock [15] have already been validated, it is known that the different epigenetic pattern among cells underlies a specific epigenetic profile of tissues or cellular populations [36]. Some of the genes proved to be tissue-specific (FHL2 for saliva and blood, TRIM59 for blood and buccal swab) [37] while accuracy of others (such as ELOVL2) results independently from the type of biological sample [38]. Up to now, the research has focused mainly on tissues that are easy to reach (blood, buccal swab, saliva, sperm). In fact, even if epigenetic clock revealed the synchronicity of DNAm age across several tissues, some exceptions have already been detected (female breast epigenetically older and cerebellum younger) [15,39]. Of utmost relevance here is that Morin et al. observed a remarkably younger age in cumulus cells when compared to peripheral cells [31]. Indeed, Hanson et al. confirmed this result in a larger cohort of infertile women undergoing ovarian stimulation [40]. However, while Morin failed to prove the association of different epigenetic ages of leukocytes to ovarian response [31], in a subgroup analysis Hanson et al. observed that white blood cells were epigenetically older in poor ovarian responders [40]. Even if these observations may be of scant interest in disentangling predictors of success (cumulus cells can be obtained only when performing IVF), studying more in-depth aging of ovarian cells could reveal new avenues of research. In line, Olsen et al. have recently developed a "Granulosa Cell clock" [33] apparently more accurate for mural granulosa cells than *Horvath's clock* [15] and *Skin & Blood clock* [41]. These results could be a promising confirm of the distinct methylation trajectory with age of follicle cells compared to other somatic cells. Moreover, epigenetic profile changes have been defined in granulosa cells in women with different ovarian reserve, suggesting the possibility of a methylation DNA variability in relation to granulosa cells functionality [42].

Some limitations of the study should be recognized. Firstly, other methodological approaches could be used to assess mt-DNA, TL and DNAm. As alluded above, this limitation could be particularly important for DNAm age. On the other hand, it should be underlined that the methods we adopted were already validated by others [43]. In addition, one has to point out that available scientific evidence is insufficient to definitely indicate a referral method [44]. Secondly, infertile patients were enrolled irrespective of infertility diagnosis. This could have diluted the results and precluded meaningful subgroup analyses. Thirdly, we decided to exclusively focus on women aged 37–39 years and conclusions should thus be considered valid only for this age group. This decision was taken because we believed that the evaluation of biological age could be particularly relevant in women in their late thirties, when natural fertility is rapidly declining. Evidence from less selected women is however required to better disentangle the possible clinical utility of assessing biological age. Finally, any deduction from our subgroup analysis may be limited by the reduced sample size in our research. Nevertheless, results obtained in endometriosis group are extremely intriguing and more convincing evidence is warranted through wider research in this field.

In conclusion, peripheral assessment of mt-DNA, LINE-1 methylation, TL and DNAm age do not provide a potent mean to predict the success of IVF. However, our findings strengthen the interest of investigating more in-depth the possible role of DNAm age. Along with the state of research, our study highlights the urgency to shed light on the epigenetic mechanisms of female infertility. Up to now, DNAm age is neither accurate or powerful enough for a clinical purpose. However, the development of alternative and more precise epigenetic clocks for different cellular populations (i.e., granulosa, cumulus cells) could open new fruitful avenues of research, providing new potential markers not only to predict IVF success but also to streamline the selection of the "eligible woman" to ART treatment.

## Supporting information

**S1 Table. Biomarkers of aging in women who did and did not achieve a live birth according to the IVF indication.**
(DOCX)

**S1 File. Data are reported as mean ± SD; p: p-value; DNAm Age: Biological age.**
(XLSX)

## Author Contributions

**Conceptualization:** Letizia Li Piani, Edgardo Somigliana, Valentina Bollati.

**Data curation:** Letizia Li Piani, Marco Reschini, Andrea Busnelli, Chiara Favero, Benedetta Albetti, Mirjam Hoxha.

**Formal analysis:** Letizia Li Piani, Marco Reschini, Chiara Favero.

**Funding acquisition:** Edgardo Somigliana.

**Investigation:** Edgardo Somigliana, Stefania Ferrari, Benedetta Albetti, Mirjam Hoxha.

**Methodology:** Marco Reschini, Edgardo Somigliana, Paola Viganò, Benedetta Albetti, Mirjam Hoxha, Valentina Bollati.

**Project administration:** Edgardo Somigliana.

**Resources:** Stefania Ferrari, Valentina Bollati.

**Supervision:** Edgardo Somigliana, Paola Viganò.

**Validation:** Edgardo Somigliana, Paola Viganò.

**Visualization:** Valentina Bollati.

**Writing – original draft:** Letizia Li Piani, Edgardo Somigliana.

**Writing – review & editing:** Letizia Li Piani, Edgardo Somigliana, Andrea Busnelli, Paola Viganò, Valentina Bollati.

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
