## [Decision Letter · Decision Letter 0]

27 Oct 2021

PONE-D-21-26763Peripheral mitochondrial DNA, telomere length and DNA methylation as predictors of live birth in  in vitro  fertilization cyclesPLOS ONE

Dear Dr. Li Piani,

Thank you for submitting your manuscript to PLOS ONE. After careful consideration, we feel that it has merit but does not fully meet PLOS ONE’s publication criteria as it currently stands. Therefore, we invite you to submit a revised version of the manuscript that addresses the points raised during the review process.

We look forward to receiving your revised manuscript.

Kind regards,

Gabriele Saretzki, PhD

Academic Editor

PLOS ONE

Journal Requirements:

“I have read the journal's policy and the authors of this manuscript have the following competing interests:

Dr. Somigliana reports grants from Ferring, grants and personal fees from Merck-Serono, grants and personal fees from Theramex, personal fees from Gedeon-Richter, outside the submitted work.”

Additional Editor Comments (if provided):

Please address the issues of the 2 reviewers.

Reviewers' comments:

Reviewer's Responses to Questions

**Comments to the Author**

1. Is the manuscript technically sound, and do the data support the conclusions?

Reviewer #1: Yes

Reviewer #2: Partly

2. Has the statistical analysis been performed appropriately and rigorously? 

Reviewer #1: Yes

Reviewer #2: Yes

3. Have the authors made all data underlying the findings in their manuscript fully available?

Reviewer #1: Yes

Reviewer #2: Yes

4. Is the manuscript presented in an intelligible fashion and written in standard English?

Reviewer #1: Yes

Reviewer #2: Yes

5. Review Comments to the Author

Reviewer #1: This is an interesting study that has investigated peripheral genetic markers and corelated them prospectively with live birth rates in IVF. The results are, as the authors conclude, of no practical value in counselling patients about their potential IVF success. Nonetheless, they add to the scientific knowledge about ovarian ageing which is important for human fertility.

The results of the study are not entirely novel but adds to the growing number of investigations in this field. The negative corelations are important but the results in Table 4 related to Biological age are more interesting. The mechanism of ovarian ageing related to oocyte numbers and aneuploidy should probably remain the main focus of scientific study but this paper links it to more general ageing processes. As such, the corelation is relevant.

Live birth rate has been selected as the primary outcome and this is understandable as it is the most important clinical parameter for patients. To investigate the underlying biological process, it might have been more informative to consider the embryological stages e.g. fertilisation rates (what were the chronological and biological ages of the men?), embryo development rates (at what point was development compromised?)

The data in Tables 1 and 2 is consistent with previously published and well accepted parameters related to IVF outcome.

Of historical interest: old teachings were that “a woman with a wrinkled face would have wrinkled ovaries irrespective of age”. We are now confirming this more general association with biological markers.

Specific points:

It would be helpful to use the terms either DNAm age or Biological age consistently in the paper. For someone not familiar with the terms, it is confusing.

The references are not numbered in the Reference section.

There are several grammatical corrections needed.

Table 3 – is the data for the OR for LINE-1 and Biological age correct?

Reviewer #2: The topic of the work is very interesting, and the paper is well written. However, authors should make subgroups of women according to their pathology to analyse whether age biomarkers may give different results depending on each pathology.

6. PLOS authors have the option to publish the peer review history of their article (what does this mean?). If published, this will include your full peer review and any attached files.

Reviewer #1: **Yes: **Alison Murdoch

Reviewer #2: **Yes: **María Elisa Varela Varela Sanz

---

## [Author Response · Author response to Decision Letter 0]

17 Nov 2021

Response to Reviewers

Reviewer #1: This is an interesting study that has investigated peripheral genetic markers and corelated them prospectively with live birth rates in IVF. The results are, as the authors conclude, of no practical value in counselling patients about their potential IVF success. Nonetheless, they add to the scientific knowledge about ovarian ageing which is important for human fertility.

The results of the study are not entirely novel but adds to the growing number of investigations in this field. The negative corelations are important but the results in Table 4 related to Biological age are more interesting. The mechanism of ovarian ageing related to oocyte numbers and aneuploidy should probably remain the main focus of scientific study but this paper links it to more general ageing processes. As such, the corelation is relevant.

We would like to thank you the Reviewer for the comment.

Live birth rate has been selected as the primary outcome and this is understandable as it is the most important clinical parameter for patients. To investigate the underlying biological process, it might have been more informative to consider the embryological stages e.g. fertilisation rates (what were the chronological and biological ages of the men?), embryo development rates (at what point was development compromised?)

We would like to thank you the Reviewer for the comment. Although the observation of the Reviewer is interesting, we deem important to be able to predict live birth rather than embryological parameters. It should be noted that, giving more attention to secondary outcomes rather than to the main clinical endopoint may lead to uncorrect conclusions with possible negative consequences.

The data in Tables 1 and 2 is consistent with previously published and well accepted parameters related to IVF outcome. Of historical interest: old teachings were that “a woman with a wrinkled face would have wrinkled ovaries irrespective of age”. We are now confirming this more general association with biological markers.

We would like to thank you the Reviewer for the comment.

Specific point: It would be helpful to use the terms either DNAm age or Biological age consistently in the paper. For someone not familiar with the terms, it is confusing.

The text has been corrected accordingly.

The references are not numbered in the Reference section.

The text has been corrected accordingly.

There are several grammatical corrections needed.

The text has been corrected by an English speaker.

Table 3 – is the data for the OR for LINE-1 and Biological age correct?

We would like to thank the Reviewer for the comment. Table 3 shows data as mean values while Table 4 shows OR. Therefore, lower panel of Table 3 has been cancelled.

This work aims to analyse the potential of different age-indicators (telomere length, mitochondrial DNA and DNA methylation pattern) as possible markers of success after IVF treatments. The topic is of interest for the reproductive field. The manuscript is well-written, and authors provide key references to the whole comprehension of the work.

We would like to thank you the Reviewer for the comment.

 However, there are several points that could be improved:

Major points:

-The authors recruited a total of 181 patients for IVF or ICSI. These people had different pathologies: endometriosis, anovulatory problems, tubal factor, even the male factor could be affecting the results. It is possible that the imprint of aging changes in a different manner in different pathologies. In addition, it does not seem compatible to find a common pattern between women with a specific reproductive pathology and women undergoing IVF because of male factor.

Making subgroups of patients, according to each pathology would help to understand whether the analysed biomarkers (telomere length, mitochondrial DNA and DNA methylation pattern) do work for any given pathology, rather than for all of them together. It would be better to compare the results of age markers of each subgroup/pathology with healthy controls of similar age. 

We thank the Reviewer for the comment. Following Reviewer’s advice, we performed a subgroup analysis based on the leading cause of infertility. We found out that only two groups (endometriosis and mixed factor) were characterized by a statistically significant difference of biological age and age acceleration between women who delivered and who did not. While we cannot draw any conclusions in case of mixed factor both for the heterogeneity of the population and the reduced sample size, the result related to endometriosis is intriguing. Besides, this data was confirmed even comparing endometriosis to unexplained infertile women, that could be considered the control group. It could be hypothesized that endometriosis may accelerate ovarian aging more than expected. The data and the relative text in the Results and Discussion has been included in the manuscript (Supplementary table).

-Line 232 and 233: “The contrast of our findings with those reported by Busnelli et al. is more difficult to reconcile”. Could you please try to give a possible explanation for these differences, for example referring to the limitations of either your or their study?

We thank the Reviewer for the comment, the sentence was better clarified introducing some explanations for these differences.

-Please explain the meaning of your results to facilitate the reading of the manuscript. For example: Odds ratio of 0.9 (in the case of mDNA) would indicate that the probability of having a live newborn is lower each time your DNA methylation fingerprint corresponds to a year older.

We thank the Reviewer for the comment. This aspect has now been more thoroughly discussed in the text. References have been added.

-Table 2. The number of oocytes retrieved, and the number of cleavage stage embryos is repeated twice with different values. Please explain better in the text or below the table what is the difference between both results. It is really not easy to understand.

We thank the Reviewer for the comment. In regard to oocytes, “ N. of oocytes retrieved” refers to the average value of oocytes after the retrieval, while “No oocytes retrieved” refers to the number of women without oocytes retrieved. We have added this explanation below the table, as suggested, to make this difference clearer. In regard to “number of cleavage stage embryos” we wrongly expressed the same value in mean and median. We have corrected this mistake. 

Minor points:

-Line 65 and Line 72: “Aging” and “ageing” should be unified. 

Corrected

-Line 122: “sample” should be replaced by “samples”. 

Corrected

-Line 245: “Leucocytes” should be replaced by “Leukocytes”.

Corrected

-Line 243, 247 and 249: “telomere length” should be replaced by “TL”.

Corrected

-Line 202 and onwards: The type of statistical analysis performed should be explained in materials and methods, and only a brief reference could be done in results if necessary.

We thank the Reviewer for the comment, the sentence was modified as requested.

-Since the manuscript body refers to the DNAm age when speaking of “Biological age (years)”, please explain write underneath the table that equivalence because it is confusing. 

The text has been corrected accordingly.

Reviewer #2: The topic of the work is very interesting, and the paper is well written. However, authors should make subgroups of women according to their pathology to analyse whether age biomarkers may give different results depending on each pathology.

We thank the Reviewer for the comment. Following Reviewer’s advice, we performed a subgroup analysis based on the leading cause of infertility. We found out that only two groups (endometriosis and mixed factor) were characterized by a statistically significant difference of biological age and age acceleration between women who delivered and who did not. While we cannot draw any conclusions in case of mixed factor both for the heterogeneity of the population and the reduced sample size, the result related to endometriosis is intriguing. Besides, this data was confirmed even comparing endometriosis to unexplained infertile women, that could be considered the control group. It could be hypothesized that endometriosis may accelerate ovarian aging more than expected. The data and the relative text in the Results and Discussion has been included in the manuscript (Supplementary table).

---

## [Decision Letter · Decision Letter 1]

22 Nov 2021

PONE-D-21-26763R1Peripheral mitochondrial DNA, telomere length and DNA methylation as predictors of live birth in  in vitro  fertilization cyclesPLOS ONE

Dear Dr. Li Piani,

Thank you for submitting your manuscript to PLOS ONE. After careful consideration, we feel that it has merit but does not fully meet PLOS ONE’s publication criteria as it currently stands. Therefore, we invite you to submit a revised version of the manuscript that addresses the points raised during the review process.

ACADEMIC EDITOR: One of the previous reviewers still requires some minor revision.==============================

We look forward to receiving your revised manuscript.

Kind regards,

Gabriele Saretzki, PhD

Academic Editor

PLOS ONE

Journal Requirements:

Additional Editor Comments:

One of the previous reviewers still requires some minor revision.

Reviewers' comments:

Reviewer's Responses to Questions

**Comments to the Author**

1. If the authors have adequately addressed your comments raised in a previous round of review and you feel that this manuscript is now acceptable for publication, you may indicate that here to bypass the “Comments to the Author” section, enter your conflict of interest statement in the “Confidential to Editor” section, and submit your "Accept" recommendation.

Reviewer #1: (No Response)

Reviewer #2: All comments have been addressed

2. Is the manuscript technically sound, and do the data support the conclusions?

Reviewer #1: Yes

Reviewer #2: Yes

3. Has the statistical analysis been performed appropriately and rigorously? 

Reviewer #1: Yes

Reviewer #2: Yes

4. Have the authors made all data underlying the findings in their manuscript fully available?

Reviewer #1: Yes

Reviewer #2: Yes

5. Is the manuscript presented in an intelligible fashion and written in standard English?

Reviewer #1: Yes

Reviewer #2: Yes

6. Review Comments to the Author

Reviewer #1: The short title of the paper overemphasises the results of the paper. The authors state that the association is “insufficient to claim a clinical use”. The paper is an interesting indicator for further research to understand the genetic basis of oocyte aging rather than clinical testing for treatment. Line 339: this final clause also still implies the predictive potential for DNAm. Knowing how the IVF sector responds to such comments (another expensive unhelpful test for the patient) it would be ethically more acceptable to be cautious. In practice, the predictive power would need to be many times more powerful to alter patient selection.

Abstract/ Objective Line 27.

The brackets around (biological age…rate) are confusing implying that these are the factors that are part of DNAm calculation. Line 142 equates Biological age and DNAm age so why not use just one term? The paper is written assuming that the main readers will be those in the field of Reproductive medicine/science so it is better to write to their understanding.

Reviewer #2: Reviewer 2 is satisfied with the response of the authors. Some of the points raised by reviewer 2 were written under “reviewer 1”. However, reviewer 2 has checked all the comments that were raised by Reviewer 2 from the beginning.

7. PLOS authors have the option to publish the peer review history of their article (what does this mean?). If published, this will include your full peer review and any attached files.

Reviewer #1: **Yes: **Alison Murdoch

Reviewer #2: No

---

## [Author Response · Author response to Decision Letter 1]

26 Nov 2021

Response to Reviewer #1

The short title of the paper overemphasises the results of the paper. The authors state that the association is “insufficient to claim a clinical use”. The paper is an interesting indicator for further research to understand the genetic basis of oocyte aging rather than clinical testing for treatment. Line 339: this final clause also still implies the predictive potential for DNAm. Knowing how the IVF sector responds to such comments (another expensive unhelpful test for the patient) it would be ethically more acceptable to be cautious. In practice, the predictive power would need to be many times more powerful to alter patient selection.

We thank the Reviewer for the comment. We completely agree that DNAm age should be much more powerful to be considered useful for a clinical purpose. However, in our conclusion we wanted only to emphasize the urgency of exploring ovarian DNAm aging mechanisms, not inviting to adopt it clinically. As short title could be misleading, we modified it. Similarly, we added a sentence to clarify the message before line 339.

Abstract/ Objective Line 27.The brackets around (biological age…rate) are confusing implying that these are the factors that are part of DNAm calculation. 

We thank the Reviewer for the comment, the sentence was modified as requested.

Line 142 equates Biological age and DNAm age so why not use just one term? The paper is written assuming that the main readers will be those in the field of Reproductive medicine/science so it is better to write to their understanding.

Corrected.

Reviewer #2: Reviewer 2 is satisfied with the response of the authors. Some of the points raised by reviewer 2 were written under “reviewer 1”. However, reviewer 2 has checked all the comments that were raised by Reviewer 2 from the beginning.

We would like to thank the Reviewer for the comment.

---

## [Decision Letter · Decision Letter 2]

6 Dec 2021

Peripheral mitochondrial DNA, telomere length and DNA methylation as predictors of live birth in  in vitro  fertilization cycles

PONE-D-21-26763R2

Dear Dr. Li Piani,

We’re pleased to inform you that your manuscript has been judged scientifically suitable for publication and will be formally accepted for publication once it meets all outstanding technical requirements.

Kind regards,

Gabriele Saretzki, PhD

Academic Editor

PLOS ONE

Additional Editor Comments (optional):

All outstanding issues have been addressed now.

Reviewers' comments:

Reviewer's Responses to Questions

**Comments to the Author**

1. If the authors have adequately addressed your comments raised in a previous round of review and you feel that this manuscript is now acceptable for publication, you may indicate that here to bypass the “Comments to the Author” section, enter your conflict of interest statement in the “Confidential to Editor” section, and submit your "Accept" recommendation.

Reviewer #1: All comments have been addressed

2. Is the manuscript technically sound, and do the data support the conclusions?

Reviewer #1: Yes

3. Has the statistical analysis been performed appropriately and rigorously? 

Reviewer #1: Yes

4. Have the authors made all data underlying the findings in their manuscript fully available?

Reviewer #1: Yes

5. Is the manuscript presented in an intelligible fashion and written in standard English?

Reviewer #1: Yes

6. Review Comments to the Author

Reviewer #1: Thanks for agreeing the changes and good luck with further basic science studies. There is a lot to learn in this area.

7. PLOS authors have the option to publish the peer review history of their article (what does this mean?). If published, this will include your full peer review and any attached files.

Reviewer #1: **Yes: **Professor Alison Murdoch

---

## [Editor Report · Acceptance letter]

14 Jan 2022

PONE-D-21-26763R2 

Peripheral mitochondrial DNA, telomere length and DNA methylation as predictors of live birth in *in vitro* fertilization cycles 

Dear Dr. Li Piani:

I'm pleased to inform you that your manuscript has been deemed suitable for publication in PLOS ONE. Congratulations! Your manuscript is now with our production department. 

Kind regards, 

on behalf of

Dr. Gabriele Saretzki 

Academic Editor

PLOS ONE